# Usefulness of serial post-systolic shortening by speckle tracking echocardiography to predict major adverse cardiovascular events and segmental function improvement after acute myocardial infarction

Ju-Feng Hsiao[1,☯], Kuo-Li Pan[2☯], Chi-Ming Chu[3], Shih-Tai Chang[2], Chang-Min Chung[2], Jen-Te Hsu[1] *

1 Cardiovascular Division, Luodong Poh-ai Hospital, Yilan County, Taiwan, 2 The Department of Cardiology, Chiayi Chang Gung Memorial Hospital, Chang Gung University College of Medicine, Chiayi County, Taiwan, 3 Department of Epidemiology, Section of Biostatistics and Bioinformatics, School of Public Health National Defense Medical Center, Taipei, Taiwan

☯ These authors contributed equally to this work.
* hsujente@gmail.com

## Abstract

### Purpose

The aim is to determine whether serial post-systolic shortening (PSS) using speckle tracking echocardiography (STE) could predict major adverse cardiovascular events (MACE), especially symptom-driven infarct-related artery (IRA) revascularization and improvement in segmental function in post-myocardial infarction patients.

### Methods/Results

Ninety-four patients (average age 61.1 ± 12.5 y, 84 [84.9%] male) with new-onset acute myocardial infarction were enrolled. Serial echocardiography was performed during the initial presentation, and at 3, 6 and 12 months after admission. PSS, strain and systolic strain rate were calculated using STE. Improvement in segmental function was defined as a decrease of ≧1 grade in wall motion score. During the follow-up (29.4 ± 12.7months), 22 patients (23.4%) had MACE and 17 patients had symptom-driven IRA revascularization. In multivariate model, PSS at 3 months was independently predictive for symptom-driven IRA revascularization (Hazard ratio (HR) = 0.5, 95% CI = 0.26–0.97) and for MACE (HR = 0.4, 95% CI = 0.24–0.67) (p < 0.05). Segmental function improvements were found in 255 segments (66.1%) and ROC curve analyses showed that AUC (95% CI) of the initial PSS was 0.7(0.65–0.77) (cut-off values = -1.08, sensitivity = 58%, specificity = 73% specificity).

### Conclusions

Post-systolic shortening at 3 months is an independent predictor for symptom-driven IRA revascularization and MACE. Regional wall motion recovery also could be predicted by

**Data Availability Statement:** All relevant data are within the manuscript and its Supporting Information files.

**Funding:** This study is supported by grant CMRPG6E0341 from Chang Gung Memorial hospital.

**Competing interests:** The authors have declared that no competing interests exist.

initial PSS. Serial assessment of two-dimensional STE should be investigated in post-myocardial infarction patients in the future.

## Introduction

Early risk assessment is important for patients with post-myocardial infarction, due to the risk of major adverse cardiovascular events (MACE) and mortality even after percutaneous coronary intervention (PCI). Among non-invasive tests, echocardiography is the first choice for assessing heart function and prognosis [1, 2]. Note that two-dimensional speckle tracking echocardiography (2D STE) has proven superior to conventional echocardiography to provide more information and risk stratification [3]. However, previous studies have reported that the diagnostic accuracy of 2D STE for restenosis is lower in territories of infarction [4–6]. Post-systolic shortening (PSS) is defined as persistent shortening beyond aortic valve closure. It has been observed in cases of acute and chronic ischemic diseases and has been reported as a predictor of viability [7–9] or ischemia memory which could persist longer than the decrease in peak systolic strain [10]. However, relatively few studies have evaluated the prognostic value of PSS in MACE, and particularly infarct-related artery (IRA) revascularization among patients with post-myocardial infarction. Our objective in the current study was to assess serial PSS using 2D STE and thereby determine its value in predicting MACE after myocardial infarction and particularly symptom-driven IRA revascularization. We also evaluated its capacity to predict improvement in segmental myocardial function.

## Methods

### Study population

This study enrolled patients admitted for new-onset acute myocardial infarction between March 2010 and July 2014. Patients with severe valvular disease, atrial fibrillation or flutter, and/or a history of myocardial infarction and cases with poor echocardiographic images were excluded. Fig 1 is the flow chart of the study protocol. Percutaneous coronary intervention was performed as early as possible after acute myocardial infarction was diagnosed. Serial echocardiography was performed after admission (average of 3.3 days), and at 3, 6, and 12 months after PCI. Biochemical tests included serum creatinine, high-sensitivity C-reactive protein (hs-CRP), and brain natriuretic peptide. Serial creatine kinase MB isoenzyme (CK-MB) levels were immediately collected on admission and at 8 and 16 hours later after PCI. Estimated glomerular rate (eGFR) was measured using the 4-variable Modification of Diet in Renal Disease formula. This study was approved by the Ethics Committee of the Chiayi Chang Gung Memorial Hospital, and written informed consent was provided by all patients.

### Angioplasty protocols

Once acute myocardial infarction was diagnosed, PCI was completed as soon as possible. For ST-segment elevation myocardial infarction patients, door-to-balloon time was reduced to less than 90 minutes. PCI was considered successful if the residual stenosis was < 30% and the flow after the procedure was better than thrombolysis in myocardial infarction (TIMI) grade 2. All patients received dual antiplatelet therapy with a loading dose of aspirin 100 mg and clopidogrel 300 mg and a maintenance dose of aspirin 100 mg and clopidogrel 75 mg per day. All patient underwent PCI and success rate was 100% and TIMI flow grade was 3 at the end of the procedure.

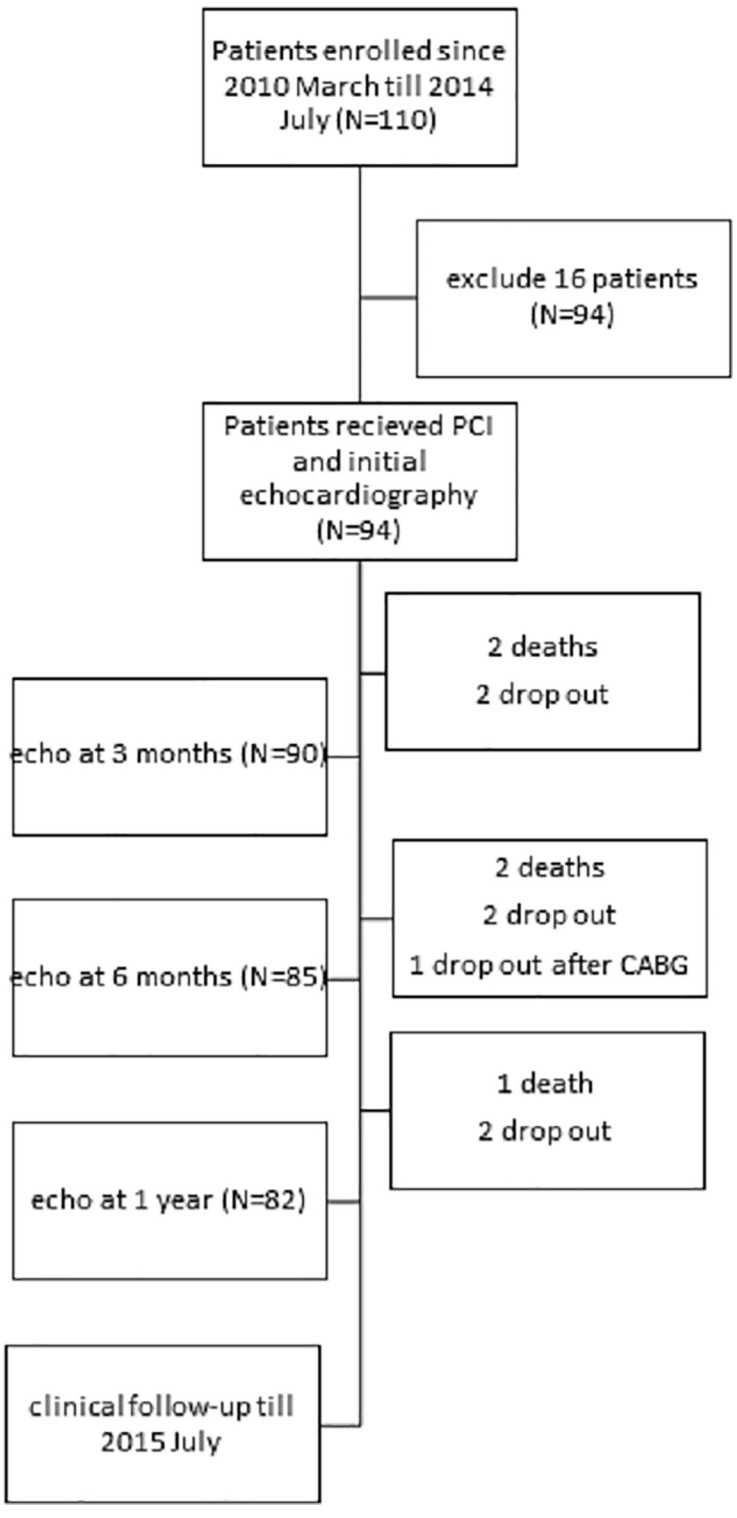

**Fig 1. Flow chart of the study protocol.**

Recorded findings of coronary angiography included the culprit vessels, diseased vessels, left main involvement, and single and multi-vessels (2 vessels). Culprit vessel was defined as infarct related artery. A diseased vessel was defined as co-morbid vessels with 50% stenosis and not related to myocardial infarction.

## Echocardiography

Comprehensive 2D transthoracic grayscale echocardiography was performed using a GE Vivid 7 echocardiographic system (M3S probe, Vivid 7, GE Vingmed, Horten, Norway) in accordance with the recommendations of the American Society of Echocardiography and the European Association of Cardiovascular Imaging [11, 12]. LVEF and LV volume were calculated using the modified Simpson's biplane method. Regional wall motion was visually evaluated using a 16-segment model with each segment scored as follows: 1 = normal, 2 = hypokinesia, 3 = akinesia, 4 = dyskinesia, and 5 = aneurysmal change. WMSI was averaged from scores of all evaluated segments (WMSI = sum of the scores of all evaluated segments / number of the evaluated segments). An improvement in segmental function was defined as an improvement in segmental wall motion of at least one grade at the 12-month follow-up. Peak early (E) and late diastolic wave velocity (A) were measured using pulse-wave velocity and tissue Doppler imaging to measure peak early (e′) and late (a′) diastolic velocities at the mitral septal annulus.

The frame rates of these images were from 66 to 79 frames/s. Images of 3 consecutive cardiac cycles in apical 2-, 3-, and 4-chamber views and short-axis views at the apex, middle, and base of the LV were stored digitally for off-line analysis using EchoPAC version 11.0 (GE Vingmed). We traced the endocardial border manually at end-systole and adjusted the width of the region of interest to cover the entire myocardium, whereupon the software automatically tracked the myocardium. Poor tracking quality was revised manually until the quality was acceptable. The end of systole was defined as the first frame when the aortic valve was closed in the apical long-axis view. Each apical view or short-axis view was divided into 6 segments. The peak longitudinal strain (LS), peak circumferential strain (CS), and radial strain (RS) and systolic longitudinal strain rate (LSRs), circumferential strain rate (CSRs), and radial strain rate (RSRs) were averaged from all 18 segments. Post-systolic shortening (PSS) was defined as persistent shortening beyond aortic valve closure (PSS = peak negative strain in the cardiac cycle—peak negative strain in systole) (Fig 2). Thus, PSS were negative values. PSS was average of segmental PSS from 18 segments. Mechanical dispersion was measured as standard deviation (SD) of time from onset Q in electrocardiography to peak negative longitudinal strain from 18 segments. Culprit longitudinal strain (culprit LS) was defined as the average value of longitudinal strain in territories of the culprit vessels, based on a standardized model of myocardial perfusion in the left anterior descending (LAD), left circumflex (LCX), and right coronary artery (RCA). Note that the strain and strain rate were calculated based on an 18-segment left ventricular model, whereas the wall motion score index was calculated based on a 16-segment model. When comparing the improvement in segmental myocardial function, the 16-segment model was used for the strain analysis by dividing the strain value at the apex into 4 segments. The two segmental strain values were averaged for apical anterior and apical lateral segments.

## Follow-up

All patients underwent follow-up examinations at clinics, and their hospital records were systematically reviewed for information related to adverse events. Major adverse cardiovascular events included all causes of death, hospitalization for heart failure, symptom-driven IRA

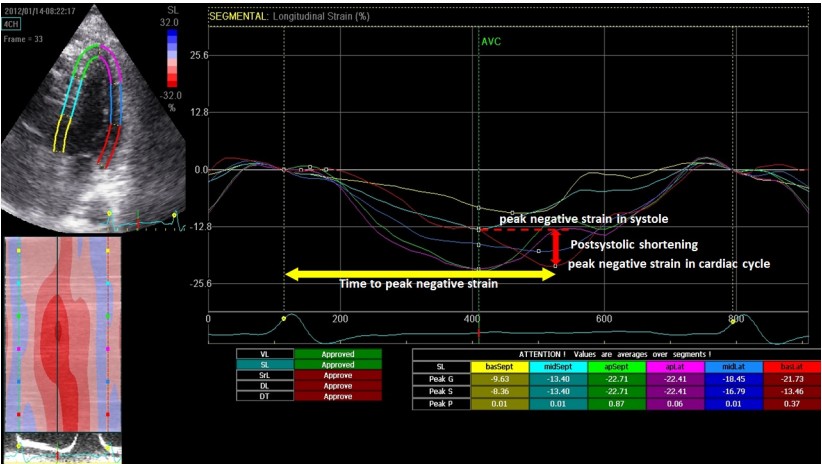

**Fig 2. Illustration of Post-systolic shortening (PSS) and time to peak negative strain.** Post-systolic shortening was defined as peak negative strain in cardiac cycle—peak negative strain in systole.

revascularization, and stroke. Hospitalization for heart failure was defined as dyspnea with chest radiographic findings of pulmonary congestion or edema involving the use of intravenous diuretics. Symptom-driven IRA revascularization was performed in cases of stenosis exceeding 70%, including *de novo* and in-stent restenosis lesions.

## Statistical analysis

Continuous variables are expressed as means ± standard deviations for normally-distributed variables and as the median (25th, 75th percentile) for variables with non-normal distribution using the Kolmogorov–Smirnov test. Categorical variables are expressed as the number of subjects and percentages. A two-tailed Student's t-test was used for continuous normally-distributed variables, the Mann-Whitney U test was used for non-normally-distributed variables, and chi-squared tests were used for categorical variables in comparing event and event-free groups. The univariate Cox proportional hazard regression model was used to evaluate associations with adverse events, wherein significant variables were entered into multivariate analysis using the forward stepwise method. The study population was divided at the mean value. Mixed model analysis was used to compare patients with and without IRA revascularization in terms of changes in conventional echocardiographic measurements or 2D STE analysis across four time-points. Each visit was treated as a categorical variable. The restricted maximum likelihood (REML) procedure was used to model parameter estimates. Aikake information criterion (AIC) and Bayesian information criterion (BIC) were used to compare and select significant models. In the assessment of improvements in segmental myocardial function, a two-tailed Student's t-test was used to compare strain and strain rates between segments with and without functional recovery. Receiver operating characteristic curves (ROC) were used to identify the optimal cut-off value, sensitivity, and specificity using the Youden index. All statistical analysis was performed using SPSS 21 (IBM SPSS Statistics for Windows, Version 21.0. Armonk, NY: IBM Corp). All tests were two-sided and a P value of $< 0.05$ was regarded as significant.

Ten subjects were selected at random to assess inter- and intra-observer variability. All LV deformation indices were measured using two independent observers for inter-observer variability. Measurements were taken again a month later for the same observers for intra-observer variability. Variability was expressed using percentages derived from the absolute difference

between the two sets of measurements, divided by the overall mean of the two sets of measurements.

## Results

Ninety-four patients (average age 61.1 ± 12.5 y, 84 [84.9%] male) were enrolled. The mean follow-up duration was 29.4 ± 12.7 months. During the follow-up, 22 patients (23.4%) reached one or more of the endpoints: 7 patients expired, 3 patients had strokes, 1 patient was admitted for heart failure, and 17 patients experienced symptom-driven IRA revascularization. The median time for MACE was 7.5 months after acute myocardial infarction. Table 1 shows clinical characteristics of the patients. Medication at discharge were as follows: aspirin 92 (97.9%), clopidogrel 89 (94.7%), angiotensin-converting enzyme (ACE) inhibitor or angiotensin receptor blocker (ARB) 41 (43.6%) and beta-blocker 50 (53.2%). Table 2 shows conventional echocardiographic findings and LV deformation indices obtained using 2D STE in the event and event-free groups. The proportion of females in the event group was higher, WMSI and LS

**Table 1. Patients' clinical characteristics between event-free and event groups.**

| Variables | Event-free group (N = 72) | Event group (N = 22) | P |
|---|---|---|---|
| Age (y) | 60.7 ± 11.2 | 62.6 ± 16 | 0.53 |
| Male sex | 67(93.1%) | 17(77.3%) | 0.05* |
| hsCRP (mg/L) | 20.3(6.1, 46.6) | 23.1(5.2, 50.4) | 0.84 |
| Peak CK-MB (ng/mL) | 38.1(6.0, 156.5) | 64(6.0,273.1) | 0.29 |
| brain natriuretic peptide (ng/L) | 153(65, 349) | 251(170, 631) | 0.06 |
| Hypertension | 43(59.7%) | 14(63.6%) | 0.74 |
| Diabetes mellitus | 20(27.8%) | 7(31.8%) | 0.71 |
| Smoking | 44(61.1%) | 11(50%) | 0.36 |
| eGFR (mL/min/1.73 m$^2$) | 76.4 ± 22.6 | 85.2 ± 32.4 | 0.24 |
| STEMI | 42(58.3%) | 17(77.3%) | 0.13 |
| D-to-B (min) | | | |
| STEMI | 70 (56, 88) | 74 (63, 88) | 0.33 |
| NSTEMI | 1458(774, 3280) | 854(249,2124) | 0.3 |
| S-to-B (h) | | | |
| STEMI | 3.3 (2.6, 4.8) | 4.9(3, 12.2) | 0.09 |
| NSTEMI | 38.2(24.1, 75.7) | 16.6 (9.8, 41.8) | 0.19 |
| Culprit LAD | 32(44.4%) | 14(63.6%) | 0.12 |
| Coronary artery disease | | | 0.49 |
| 1-vessel disease | 22(30.6%) | 6(27.3%) | |
| 2-vessel disease | 33(45.8%) | 8(36.4%) | |
| 3-vessel disease | 17(23.6%) | 8(36.4%) | |
| Diseased site | | | |
| LM | 8(11.1%) | 1(4.5%) | 0.68 |
| LAD | 63(87.5%) | 19(86.4%) | 1.0 |
| LCX | 31(43.1%) | 11(50%) | 0.57 |
| RCA | 45(62.5%) | 16(72.7%) | 0.38 |
| Killip class>II | 13(18.1%) | 6(27.3%) | 0.35 |

* P<0.05; CK-MB = creatine kinase MB isoenzyme; D-to-B = door-to-balloon time; eGFR = estimated glomerular filtration rate; hsCRP = high-sensitivity C-reactive protein; IRA revascularization = symptom-drive revascularization for the infarct related artery; LAD = left anterior descending artery; LCX = left circumflex artery; LM = left main; NSTEMI = non–ST-segment elevation myocardial infarction; RCA = right coronary artery; S-to-B = symptom-onset-to-balloon time; STEMI = ST-segment elevation myocardial infarction

**Table 2. Traditional echocardiographic findings and left ventricular deformation indices.**

| Variables | Event-free group (N = 72) | Event group (N = 22) | P |
|---|---|---|---|
| Ejection fraction (%) | 58 ± 9 | 56 ± 10 | 0.27 |
| Left ventricular end-diastolic volume (mL) | 103 ± 26 | 113 ± 48 | 0.39 |
| Left ventricular end-systolic volume (mL) | 44 ± 15 | 51 ± 32 | 0.27 |
| Left atrial volume index (mL/m$^2$) | 33± 10 | 32 ± 12 | 0.59 |
| Mitral inflow E/A | 1.1 ± 0.5 | 1.0 ± 0.3 | 0.39 |
| E/e' | 15 ± 5 | 16 ± 7 | 0.58 |
| WMSI | 1.29 ± 0.24 | 1.41 ± 0.24 | 0.05* |
| Longitudinal strain (%) | -17.8± 3.8 | -15.6 ± 4.1 | 0.02* |
| Longitudinal systolic strain rate (s$^{-1}$) | -1.1 ± 0.23 | -1.02 ± 0.20 | 0.16 |
| Culprit longitudinal strain (%) | -16.4 ± 4.5 | -14.7 ± 4.7 | 0.12 |
| Culprit longitudinal systolic strain rate (s$^{-1}$) | -1 ± 0.26 | -0.92 ± 0.24 | 0.2 |
| Circumferential strain (%) | -17.4 ± 4.7 | -16.6 ± 3.8 | 0.47 |
| Circumferential systolic strain rate (s$^{-1}$) | -1.43 ± 0.39 | -1.42 ± 0.37 | 0.94 |
| Radial strain (%) | 35.4 ± 13 | 35.8 ± 10.4 | 0.92 |
| Radial systolic strain rate (s$^{-1}$) | 1.76 ± 0.41 | 1.85 ± 0.51 | 0.41 |
| Initial post-systolic shortening (%) | -1.23 ± 0.84 | -1.63 ± 0.98 | 0.07 |
| Post-systolic shortening at 3 months | -0.83 ± 0.6 | -1.4 ± 0.69 | <0.05* |
| Mechanical dispersion (msec) | 51.3 ± 13.4 | 57.5 ± 14.4 | 0.08 |

* P<0.05; A = late diastolic wave velocity; E = peak early wave velocity; e′ = peak early diastolic velocity at mitral septum by tissue Doppler image; WMSI = wall motion score index

were worse and PSS at 3 months was more negative. Adverse event rates in cases of STEMI and NSTEMI were 28.8% and 14.3%, respectively. In the STEMI subgroup, the door-to-balloon time was longer in the event group than in the event-free group but not significant statistically. The distributions of culprit vessels and single- or multi-vessel lesions and proportions of the LAD were similar.

Seventeen patients underwent symptom-driven IRA revascularization at a median time of 7.7 months after acute myocardial infarction during the follow-up. Fig 3 and Table 3 compare patients with revascularization and non-revascularization in terms of serial changes in EF, WMSI, and myocardial deformation index. Mixed-model analysis revealed significant changes in LS, LSRs, culprit LS, and PSS at the four time-points. Between-group differences in LS, culprit LS, and PSS were observed, but not in WMSI, EF or mechanical dispersion. There was no interaction effect of time and all indices mentioned above. LS, culprit LS, and WMSI presented significant improvements between the baseline and 3-month follow-up, and thereafter they remained steady throughout the remainder of the first year. PSS in the revascularization group was lower at 3 months. Receiver-operating characteristic curve analysis revealed that PSS at 3 months was associated with symptom-driven IRA revascularization, where the area under the curve (AUC) was 0.7 (cut-off value = -1.34, sensitivity = 0.53, specificity = 0.83, P = 0.01).

Kaplan-Meier curves for PSS at 3 months with log rank test was showed in Fig 4. The population was divided into two groups according to the optimal cut-off value. Patients with PSS at 3 months < -1.34% had 3.5-fold increase in risk of symptom-driven IRA revascularization (log-rank$\chi^2$ = 11.7, p = 0.03). The incidence of symptom-driven IRA revascularization at 3 years in these two groups was 42.9% and 12.1%, respectively.

Table 4 shows the results of univariate analyses of Cox proportional hazard regression model for MACE and symptom-driven IRA revascularization. Only gender and PSS at 3 months were significantly associated with symptom-driven IRA revascularization even in

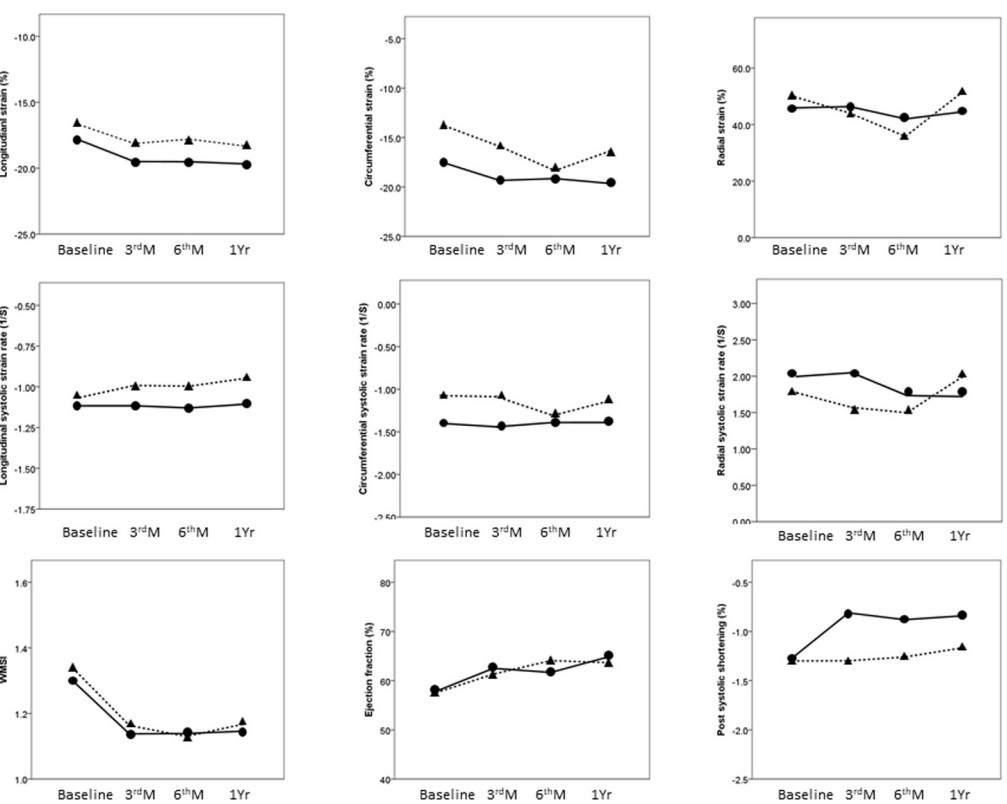

**Fig 3.** The serial changes between patients with *(dotted line)* and without *(solid line)* symptom-driven infarct related artery revascularization.

multivariate model [Hazard ratio (HR) (95% CI) for female = 3.04 (1.01–9.2), for PSS at 3 months = 0.5 (0.26–0.97), p<0.05]. Initial WMSI, EF, LS was not predictive. For MACE, gender, WMSI, LS and PSS at 3 months were significant. In the multivariate model with forward stepwise method, only PSS at 3 months was independent prognostic factors (p < 0.01, HR = 0.4, 95% CI = 0.24–0.67 for each unit increase).

## Wall motion recovery

Among the 1,504 segments from the 94 patients, there were 1039 segments where WMS = 1, 452 segments where WMS = 2, and 13 segments where WMS = 3. Table 5 lists the segmental strain and strain rates as a function of wall motion score. Myocardial deformation indices changed gradually with an increase in wall motion score. Among the segments where WMS = 1, 780 were non-culprits and 259 were culprits. Segmental LS, LSR, RS, and RSR values were worse in culprit segments, but segmental CS and CSRs were not. In both culprit and non-culprit segments, segmental LS, CS, and PSS showed improvements from the baseline.

Among the 465 segments with abnormal initial wall motion, 255 segments (66.1%) showed improvements in segmental myocardial function. Improvements from the baseline were observed in segmental LS, CS, RS, and PSS in segments where WMS = 2, but not in segments where WMS = 3. The initial segmental strain and strain rates and PSS were better in segments showing improvements in myocardial function. ROC analysis (Table 6) revealed that they were predictive of improvements in segmental myocardial function. Segmental LS had the largest area under the curve (AUC = 0.74; 95% CI, 0.68–0.79) and AUC was 0.7 (0.65–0.77) for initial PSS (cut-off value = -1.08, sensitivity = 58%, specificity = 73%).

**Table 3. Serial changes of EF, WMSI and left ventricular deformation indices between patients with symptom-driven infarct-related artery revascularization and non-revascularization.**

| | Baseline | 3rd month | 6th month | 1st year | P within groups |
|---|---|---|---|---|---|
| **Ejection fraction (%)** | | | | | 0.65 |
| non-revascularization | 57 ± 1 | 62 ± 1 | 61 ± 1 | 65 ± 1 | |
| revascularization | 56 ± 2 | 61 ± 2 | 64 ± 2 | 64 ± 2 | |
| P between two groups | 0.45 | 0.42 | 0.27 | 0.61 | |
| **WMSI** | | | | | 0.23 |
| non-revascularization | 1.31 ± 0.02 | 1.14 ± 0.03 | 1.15 ± 0.02 | 1.15 ± 0.03 | |
| revascularization | 1.38 ± 0.06 | 1.22 ± 0.06 | 1.13 ± 0.05 | 1.17 ± 0.06 | |
| P between two groups | 0.29 | 0.23 | 0.78 | 0.75 | |
| **Longitudinal strain (%)** | | | | | <0.01* |
| non-revascularization | -17.5 ± 0.5 | -19.3 ± 0.5 | -19.4 ± 0.4 | -19.7 ± 0.4 | |
| revascularization | -16 ± 0 | -17.4 ± 0.9 | -17.8 ± 0.9 | -18.3 ± 0.8 | |
| P between two groups | 0.16 | 0.08 | 0.12 | 0.16 | |
| **Longitudinal systolic strain rate ($s^{-1}$)** | | | | | <0.01* |
| non-revascularization | -1.09 ± 0.03 | -1.1 ± 0.03 | -1.11 ± 0.03 | -1.1 ± 0.03 | |
| revascularization | -1.03 ± 0.05 | -0.95 ± 06 | -1.0 ± 0.06 | -0.95 ± 0.05 | |
| P between two groups | 0.34 | 0.02* | 0.08 | 0.01* | |
| **Culprit longitudinal strain (%)** | | | | | 0.02* |
| non-revascularization | -16.2 ± 0.5 | -18.5 ± 0.5 | -18.6 ± 0.5 | -19.0 ± 0.5 | |
| revascularization | -14.8 ± 1.1 | -16.5 ± 1.07 | -17.5 ± 1.1 | -18.0 ± 1.1 | |
| P between two groups | 0.25 | 0.09 | 0.35 | 0.41 | |
| **Culprit longitudinal systolic strain rate ($s^{-1}$)** | | | | | <0.01* |
| non-revascularization | -0.99 ± 0.03 | -1.04 ± 0.03 | -1.06 ± 0.03 | -1.06 ± 0.03 | |
| revascularization | -0.94 ± 0.06 | -0.88 ± 0.06 | -0.97 ± 0.07 | -0.91 ± 0.07 | |
| P between two groups | 0.45 | 0.03* | 0.22 | 0.04* | |
| **Circumferential strain (%)** | | | | | 0.11 |
| non-revascularization | -17.2 ± 0.5 | -19.4 ± 0.6 | -19.0 ± 0.8 | -20.8 ± 0.9 | |
| revascularization | -16.9 ± 1.1 | -18.3 ± 1.1 | -17.6 ± 1.2 | -18.0 ± 1.9 | |
| P between two groups | 0.81 | 0.36 | 0.32 | 0.2 | |
| **Circumferential systolic strain rate ($s^{-1}$)** | | | | | 0.08 |
| non-revascularization | -1.39 ± 0.05 | -1.4 ± 0.04 | -1.37 ± 0.06 | -1.36 ± 0.06 | |
| revascularization | -1.31 ± 0.1 | -1.27 ± 0.08 | -1.41 ± 0.1 | -1.14 ± 0.12 | |
| P between two groups | 0.46 | 0.16 | 0.75 | 0.11 | |
| **Radial strain (%)** | | | | | 0.92 |
| non-revascularization | 35.3 ± 1.5 | 40.5 ± 2.0 | 35.3 ± 2.3 | 43.2 ± 2.7 | |
| revascularization | 36.3 ± 3 | 38.6 ± 3.6 | 38.4 ± 4.2 | 40.5 ± 5.9 | |
| P between two groups | 0.78 | 0.66 | 0.52 | 0.69 | |
| **Radial systolic strain rate ($s^{-1}$)** | | | | | 0.6 |
| non-revascularization | 1.84 ± 0.06 | 1.86 ± 0.07 | 1.67 ± 0.08 | 1.79 ± 0.07 | |
| revascularization | 1.75 ± 0.11 | 1.64 ± 0.13 | 1.7 ± 0.14 | 1.99 ± 0.15 | |
| P between two groups | 0.48 | 0.16 | 0.85 | 0.23 | |
| **Post-systolic shortening (%)** | | | | | <0.01* |
| non-revascularization | -1.3 ± 0.1 | -0.9 ± 0.1 | -0.9 ± 0.1 | -0.9 ± 0.1 | |
| revascularization | -1.5 ± 0.2 | -1.3 ± 0.2 | -1.3 ± 0.2 | -1.2 ± 0.2 | |
| P between two groups | 0.5 | 0.01* | 0.16 | 0.19 | |
| **Mechanical dispersion (msec)** | | | | | 0.09 |
| non-revascularization | 51.8 ± 13.5 | 50.8 ± 25.1 | 48.9 ± 13.7 | 53.6 ± 28.1 | |

*(Continued)*

**Table 3.** (Continued)

| | Baseline | 3rd month | 6th month | 1st year | P within groups |
|---|---|---|---|---|---|
| revascularization | 56.9 ± 14.6 | 57.6 ± 14.2 | 52.8 ± 16.0 | 57.6 ± 24.8 | |
| P between two groups | 0.18 | 0.3 | 0.33 | 0.62 | |

* = p Vale <0.05

Intra-observer variability for LS, CS and RS was 3.4 ± 2.1%, 3.5±2.8 and 9.4±8.6, respectively. For LSRs, CSRs and RSRs, it was 6.8±7.3%, 6.4±8.3% and 11.8±5.5%. Inter-observer variability for LS, CS and RS was 7.1±5.1%, 8.5±8.9% and 11.8±6.9%. For LSRs, CSRs and RSRs, it was 10.7±5.0%, 7.7±5.8% and 12.6±10.5%.

## Discussion

In the current study, 2D STE was used to serially assess post-myocardial infarction patients for a period of one year. We discovered that PSS at 3 months was an independent predictor of MACE and symptom-driven IRA revascularization, despite the fact that EF exceeded 45% in most of the study population. Myocardial deformation indices including PSS based on 2D STE results are good predictors of improvement in segmental myocardial function at 1 year.

### Prediction of symptom-driven infarct-related artery revascularization

Stress echocardiography and myocardial perfusion imaging have been investigated to diagnose restenosis after PCI [13–15]. These two non-invasive modalities are comparable in terms of sensitivity (roughly 70%) and specificity (roughly 80%); however, the diagnostic accuracy in territories of prior myocardial infarction were lower according to the previous studies [4–6]. Isaaz [16] and Buchler [17] et al used myocardial perfusion imaging at 6 months to detect

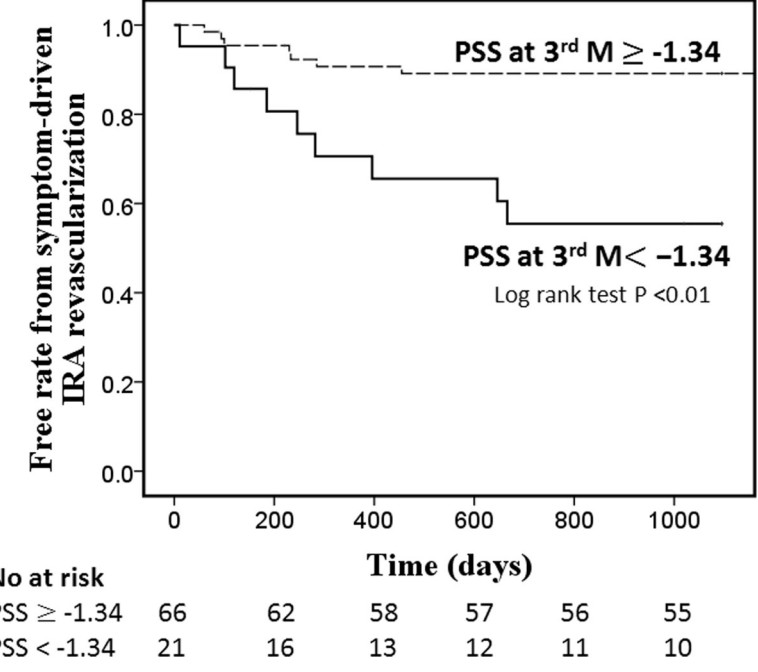

**Fig 4. Kaplan-Meier curve for PSS at 3 months for symptom-driven infarct related artery revascularization.**

**Table 4. Cox proportional hazard regression model for major adverse cardiovascular events and symptom-driven infarct related artery (IRA) revascularization.**

|  | MACE | | IRA revascularization | |
|---|---|---|---|---|
|  | Hazard ratio (95%CI) | P value | Hazard ratio (95%CI) | P value |
| male | 0.33(0.12–0.89) | 0.03* | 0.21(0.07–0.60) | <0.01* |
| Peak CK-MB(ng/mL) | 1(1–1.01) | 0.09 | 1.0(0.99–1.01) | 0.28 |
| brain natriuretic peptide (ng/L) | 1(1–1) | 0.24 | 1.0(0.99–1.0) | 0.45 |
| eGFR (mL/min/1.73m2) | 1.01(0.99–1.03) | 0.14 | 1.02(0.99–1.03) | 0.09 |
| EF | 0.97(0.93–1.02) | 0.21 | 0.97(0.92–1.02) | 0.2 |
| WMSI | 5.5(1.12–27.2) | 0.04* | 4.3(0.65–28.63) | 0.13 |
| Longitudinal strain (%) | 1.14(1.03–1.27) | 0.01* | 1.12(0.99–1.26) | 0.08 |
| Longitudinal systolic strain rate (s$^{-1}$) | 4.52(0.65–31.4) | 0.13 | 3.69(0.39–35.2) | 0.26 |
| Culprit longitudinal strain (%) | 1.09(0.99–1.19) | 0.09 | 1.08(0.97–1.21) | 0.17 |
| Culprit longitudinal systolic strain rate (s$^{-1}$) | 2.98(0.58–15.4) | 0.19 | 2.65(0.39–18.17) | 0.32 |
| Circumferential strain (%) | 1.04(0.95–1.15) | 0.42 | 1.03(0.92–1.15) | 0.6 |
| Circumferential systolic strain rate (s$^{-1}$) | 1.14(0.37–3.52) | 0.83 | 0.98(0.28–3.47) | 0.98 |
| Radial strain (%) | 1.0(0.97–1.03) | 0.95 | 0.99(0.96–1.04) | 0.96 |
| Radial systolic strain rate (s$^{-1}$) | 1.45(0.54–3.86) | 0.46 | 1.24(0.4–3.862) | 0.72 |
| Initial post-systolic shortening (%) | 0.64(0.42–1.0) | 0.05* | 0.728(0.43–1.24) | 0.24 |
| Post-systolic shortening at 3 months (%) | 0.4(0.24–0.67) | <0.05 | 0.44(0.24–0.79) | <0.01* |
| Mechanical dispersion (msec) | 1.03(0.99–1.06) | 0.1 | 1.03(0.99–1.07) | 0.18 |

* = p Vale <0.05; for acronym key, see Table 2

restenosis in STEMI patients treated via primary PCI. In those studies, the sensitivities were 48% and 54.5% and specificities were 61% and 87.8%, respectively. In the current study, PSS at 3 months and gender were the only independent factors associated with symptom-driven IRA revascularization. The AUC of ROC analysis of PSS at 3 months was 0.7 with sensitivity of 53% and specificity of 83%. Note that 2D STE is non-invasive with predictive value comparable to myocardial perfusion imaging without exposing the patient to radiation. Initial PSS was not predictive in the current study. Based on previous reports, the potential explanation is that PSS persisted even after reperfusion and is probably indicative of myocardial stunning or ischemic memory [10]. Thus, we presume that PSS at 3 months could be used to detect ongoing ischemia or restenosis rather than initial PSS.

**Table 5. Segmental strain and strain rate.**

|  | WMS = 1 | | | | WMS = 2 | | | |
|---|---|---|---|---|---|---|---|---|
|  | baseline | 3$^{rd}$ month | baseline | 3$^{rd}$ month | baseline | 3$^{rd}$ month | baseline | 3$^{rd}$ month |
|  | Non-culprit (N = 780) | | Culprit (N = 259) | | Improving (N = 247) | | Non-improving (N = 130) | |
| Segmental longitudinal strain (%) | -18.9±4.8 | -19.9±4.6$^\dagger$ | -17.6±5.7* | -19.5±5.1$^\dagger$ | -16.2±5.6 | -18.5±5.6$^\dagger$ | -11.3±4.9* | -13.8±5*$^\dagger$ |
| Segmental longitudinal systolic strain rate (s$^{-1}$) | -1.19±0.4 | -1.14±0.33$^\dagger$ | -1.06±0.4* | -1.1±0.35 | -0.99±0.36 | -1.04±0.35 | -0.71±0.33* | -0.74±0.28* |
| Segmental post systolic shortening (%) | 1.0±1.4 | 0.74±1.15$^\dagger$ | 1.2±1.6* | 0.79±1.36$^\dagger$ | 1.4±1.72 | 0.97±1.52$^\dagger$ | 2.8±2.6* | 2.02±2.2*$^\dagger$ |
|  | Non-culprit (N = 583) | | Culprit (N = 197) | | Improving (N = 183) | | Non-improving (N = 118) | |
| Segmental circumferential strain (%) | -18.0±8.5 | -19.5±9.0$^\dagger$ | -18.8±8.1 | -22.3±7.9*$^\dagger$ | -15.4±7.5 | -18.2±8.3$^\dagger$ | -11.2±7.5* | -12.9±6.7*$^\dagger$ |
| Segmental circumferential systolic strain rate (s$^{-1}$) | -1.54±0.54 | -1.47± 0.5$^\dagger$ | -1.43±0.58 | -1.52±0.44 | -1.3±0.48 | -1.38±0.49 | -1.03±0.61* | -1.09±0.44* |
| Segmental radial strain (%) | 40.9±18.4 | 44.8±19.5 | 37.7±18.8* | 40.3±20.8* | 32.6±17.7 | 39±17.6$^\dagger$ | 20.1±14.5* | 27±17.9*$^\dagger$ |
| Segmental radial systolic strain rate (s$^{-1}$) | 1.87±0.63 | 1.84±0.57 | 1.72±0.59* | 1.74±0.63 | 1.71±0.64 | 1.71±0.61 | 1.4±0.71* | 1.47±0.68* |

*P<0.05 between non-culprit vs culprit segments or improving vs non-improving segments

†P<0.05 between baseline and 3$^{rd}$ month

**Table 6. Receiver operating characteristic curve analyses for segmental function recovery.**

|  | AUC | Cut-off level | sensitivity | specificity |
|---|---|---|---|---|
| Segmental longitudinal strain (%) | 0.74(0.68–0.79) | -12.5 | 73% | 67% |
| Segmental longitudinal systolic strain rate ($s^{-1}$) | 0.73(0.67–0.78) | -0.71 | 79% | 60% |
| Segmental circumferential strain (%) | 0.67(0.59–0.74) | -16.1 | 47% | 82% |
| Segmental circumferential systolic strain rate ($s^{-1}$) | 0.65(0.58–0.73) | -0.97 | 76% | 55% |
| Segmental radial strain (%) | 0.71(0.64–0.78) | 18.6 | 78% | 59% |
| Segmental radial systolic strain rate ($s^{-1}$) | 0.62(0.55–0.70) | 1.36 | 66% | 56% |
| Segmental post-systolic shortening | 0.7(0.65–0.77) | -1.08 | 58% | 73% |

All p valve $<0.05$

## Prediction of major adverse cardiovascular events

Left ventricular systolic function (e.g., EF) and WMSI have been identified as major predictors of post-myocardial infarction outcomes [1, 18]. In 2008, Park et al. [19] were the first to report the predictive value of LS by STE for adverse events and LV remodeling. Ersboll et al. [20] reported that LS was associated with all-cause mortality and hospitalization for heart failure among patients with preserved EF (EF>40%). Antoni et al. [21] reported that global LS and LSRs were superior to LVEF and WMSI in terms of the prediction of all-cause mortality and adverse events. In the current study, multivariate Cox regression analysis identified PSS at 3 months as the only significant predictor of all-cause mortality and adverse events. PSS at 3 months was also superior to EF, WMSI, and initial LS in predicting outcomes. Hung et al. [22] reported that CS and CSRs in addition to LS and LSRs were predictive in patients at high risk of heart failure or LV dysfunction. In the current study, CS and RS were not predictive, perhaps because EF was higher than 45% in most of the patients and the circumferential and radial deformation were relatively preserved.

## Prediction of regional wall motion recovery

It might be possible to use predictions of reversible myocardial dysfunction to identify viable myocardia in cases of acute myocardial infarction. A number of studies have used STE to monitor regional wall motion recovery [23–26]. Abate et al. [25] reported that segmental LS by using 3D STE is predictive of regional wall motion recovery. Carasso et al. [23] demonstrated that LS is more sensitive to acute ischemia, and CS is more strongly correlated with segmental wall motion recovery. Orri et al. [26] reported CS can identify reversible myocardial dysfunction with accuracy comparable to that of magnetic resonance imaging. In the current study, strain and systolic strain rates in all three directions were predictive of functional recovery, and longitudinal strain presented the largest AUC (0.74). These results differ from those of Carasso and Orri, probably due to the fact that heart function was preserved in most of our patients, such that circumferential strain was not required to compensate for longitudinal dysfunction. The incidence of regional wall motion recovery (66%) was higher than in previous studies: Altiok (47%), Abate (48%), and Orii (56%) [24–26].

## Clinical implications of post-systolic shortening

Several studies have sought to determine whether PSS could be used to identify cases of acute ischemia or predict cardiac viability. Using contrast ventriculography, Hosokawa et al. [9] confirmed that PSS is indicative of viable myocardium and predictive of the recovery of systolic LV function after reperfusion. In 2011, Eek [7] used 2D STE to demonstrate that PSS is

associated with improved myocardial function after revascularization in patients with acute MI. However, other studies provided contrary findings [27–29]. Terkelsen [28] reported that PSS occurred more frequently in infarcted segments but was not associated with functional improvement. Lim et al. [27] reported that PSS may be associated with the transmural extent of necrosis in magnetic resonance imaging but did not identify it as a specific marker of viability in cases of chronic ischemia. In the current study, initial PSS was inversely proportional to wall motion scores. In segments with normal wall motion (WMS = 1), the initial PSS was more strongly negative in culprit than in non-culprit segments. In segments where WMS = 2, initial PSS was more strongly negative in non-improving segments than in improving ones. Our findings support that PSS is associated with acute ischemia and the severity of myocardial dysfunction. To the best of our knowledge, few studies have addressed the prognostic potential of PSS in patients with post-myocardial infarction after PCI. Brainin [30] was the first to report that PSS is a strong predictor of heart failure in STEMI populations after primary PCI; however, it was not identified as a predictor of death or MI. In the current study, initial PSS was predictive of improvements in segmental myocardial function (AUC = 0.7) and PSS at 3 months was associated with symptom-driven IRA revascularization and MACE. PSS is more than just ischemia memory; it has clinical implications in estimating the risk of restenosis and setting new targets for re-intervention.

## Limitations

This study was subject to a number of limitations, which should be considered in the interpretation of our results. Our study cohort was relatively small, and only a few of the patients presented adverse events. The enrolment of female patients was low (10.6% female) and the adverse events rate was higher in female than in male (50% vs 20.2%). The disproportionate number of female patients might affect the results. Patients requiring mechanical ventilation or intra-aortic balloon pumping were excluded, due to the fact that we required images with vivid endocardial edges throughout the entire cardiac cycle for 2D speckle tracking analysis. In addition, our study cohort was not limited to those with STEMI, and some patients with Killip class IV were also included. Nonetheless, our results demonstrate the feasibility of using 2D STE in real-world settings dealing with patients with acute coronary syndrome. In the current study, we performed echocardiography after PCI (average of 3.2 ± 1.6 days after admission), unlike previous studies in which echocardiography was performed on the first day of admission or immediately after PCI. Recovery of myocardial contractility following ischemia or infarction after PCI can occur early [31]. Only 43.6% the patients at discharge took ACE- inhibitor or ARB and 53.2% patients took Beta-blocker due to EF in most of the patients in this study exceeded 45%. There was no record of medication of the follow-up. Comparison of strain and strain rates between segments with and without functional recovery was not performed in segments where WMS = 3, due to the fact only one segment showed improvement. Finally, repeated coronary angiographies in the follow-up were not done for all patients, therefore, our results could not be used to determine the diagnostic accuracy of IRA restenosis

## Conclusions

Post-systolic shortening at 3 months is an independent predictor for symptom-driven IRA revascularization and MACE. Regional wall motion recovery also could be predicted by initial strain, strain rate and PSS. Serial assessment of two-dimensional STE should be investigated in post-myocardial infarction patients in the future.

## Supporting information

**S1 Data.**
(XLSX)

## Author Contributions

**Data curation:** Ju-Feng Hsiao, Kuo-Li Pan, Chi-Ming Chu, Jen-Te Hsu.

**Formal analysis:** Ju-Feng Hsiao, Kuo-Li Pan.

**Funding acquisition:** Jen-Te Hsu.

**Investigation:** Ju-Feng Hsiao, Jen-Te Hsu.

**Methodology:** Kuo-Li Pan.

**Project administration:** Jen-Te Hsu.

**Resources:** Shih-Tai Chang, Chang-Min Chung, Jen-Te Hsu.

**Software:** Ju-Feng Hsiao.

**Supervision:** Shih-Tai Chang.

**Writing – original draft:** Ju-Feng Hsiao.

**Writing – review & editing:** Ju-Feng Hsiao, Kuo-Li Pan.

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
