## [Decision Letter · Decision Letter 0]

4 Aug 2020

PONE-D-20-11858

Usefulness of serial postsystolic shortening by speckle tracking echocardiography to predict major adverse cardiovascular events and segmental function improvement after acute myocardial infarction

PLOS ONE

Dear Dr. Hsu,

Thank you for submitting your manuscript to PLOS ONE. After careful consideration, we feel that it has merit but does not fully meet PLOS ONE’s publication criteria as it currently stands. Therefore, we invite you to submit a revised version of the manuscript that addresses the points raised during the review process.

There are several concerns noted by the reviewers below. Please adequately address these points as much as possible.

We look forward to receiving your revised manuscript.

Best regards,

John Lynn Jefferies, MD, MPH, FACC, FAHA

Academic Editor

PLOS ONE

Journal Requirements:

Reviewers' comments:

Reviewer's Responses to Questions

**Comments to the Author**

1. Is the manuscript technically sound, and do the data support the conclusions?

Reviewer #1: Partly

Reviewer #2: Partly

2. Has the statistical analysis been performed appropriately and rigorously? 

Reviewer #1: Yes

Reviewer #2: Yes

3. Have the authors made all data underlying the findings in their manuscript fully available?

Reviewer #1: Yes

Reviewer #2: Yes

4. Is the manuscript presented in an intelligible fashion and written in standard English?

Reviewer #1: Yes

Reviewer #2: No

5. Review Comments to the Author

Reviewer #1: In this submission, the author tried to evaluate if the serial analysis of postsystolic shortening (PSS) by speckle tracking echocardiography (STE) could predict major adverse cardiovascular events (MACE), especially symptom-driven infarct-related artery (IRA) revascularization, and the segmental function improvement in post-myocardial infarction patients. They found that PSS is independently predictive for IRA revascularization and MACE. Initial PSS is also related to regional wall motion recovery. However, there are several issues that need to be discussed.

1. The author used PSS to evaluate the LV function for patients with newly-onset acute myocardial infarction. But it is not clear about how to measure/calculate the initial and after-admission PSS for the whole LV? Was it the average PSS for all 16 or 18 segments? Or is it the average PSS for all segments that showed postsystolic shortening? The former one may seem more reasonable. What is more, how about the time interval between AVC and PSS peak, which reflect the dyssynchrony of the LV and just like the standard deviation for time to peak of either strain or strain rate for all 16 and 18 LV segments. Time related parameters are also essential for evaluating the LV function after PCI and prognosis.

2. Other questions for PSS: the author mentioned that PSS at 3rd month is the independently predictive of IRA revascularization and MACE. And Seventeen patients received symptom-driven IRA revascularization at the median time of 7.7 months after acute myocardial infarction during the follow-up. (1) How about the time for MACE? (2) Dose all IRA revascularization after 3 months after admission? Because the result seemed to show that PSS can only indicate the IRA revascularization and MACE after 3 months after admission. Please discuss why only PSS at 3rd month is predictive of IRA revascularization and MACE?

2. The author used 16 segment model when calculated the WMSI and used 18 segment model to measure the strain and strain rate. Because the patients were enrolled from 2010 to 2013 and the 12 months follow-up echocardiography examination were ended at around the July 2014, and the echo data were all recorded, so why the author did not use 17 segment model for both WMSI and speckle tracking imaging (at least for longitudinal strain) when they analyzed the data now? They used the averaged strain and strain rate from all 18 segment, and how about the left ventricular bullseye plots? The global strain could be gotten from the bullseye plots.

3. It was a little bit hard to follow how patients had been treat and followed up. For example, when patients were admitted into the hospital, did all the patients had PCI? The author mentioned that PCI was performed as early as possible with high success rate, and information about diseased vessels were recorded, but they also mentioned that coronary angiographies were not done for all patients in limitation. The author did not mention when did they had the first echocardiography examination before “limitation”, and it should be put down in the “method”. Did every patient had been followed up for 12 months? In the method, the author said that patients were followed up for 12 months, but the results showed that the follow-up time is 29.4 ± 12.7 months. How about the dropout rate? A figure or a table to illustration the detail could be helpful.

4. More details should be given when describing the angioplasty protocols, like did every patient underwent this procedure? How did the doctor decide when and whether the patient need the PCI? What is the definition about culprit vessel? Likewise, please provide the calculation formula for WMSI, and dose IRA revascularization include PCI and CABG?

5. How about the medicine use for enrolled patients when they admitted and during the follow-up? Did the hypertension and DM under control, because they might affect the MAC and IRA revascularization too.

Minor comments:

1. Then author mentioned that they used the 2009 and 2015 recommendations of the American Society of Echocardiography and the European Association of Cardiovascular Imaging when preforming echo. Dose it means that they use 2009 guidelines when doing the conventional echocardiography analysis, while use the 2015 guidelines when measuring the STE parameter?

2. Biochemical tests had been done immediately and at 8 hours and 16 hours later. Please make it clearer about the time point, like if it had been done 8 hours and 16 hours later after admission or PCI.

3. “The end of systole was defined as the first frame of the aortic valve in apical long-axis view, and each apical view or short-axis view was divided into 6 segments”. It should be “aortic valve close”. And “pule wave Doppler” should be better than “pulse wave velocity”.

4. For SPSS 21.0, it should belong to IBM Corp (Released 2012. IBM SPSS Statistics for Windows, Version 21.0. Armonk, NY: IBM Corp).

Reviewer #2: This study sought to determine, in patients presented with acute MI who were successfully treated with PCI, whether serial measurements of postsystolic shortening (PSS) by speckle tracking echocardiography could identify individuals at increased risk of developing major adverse cardiovascular events (MACE), and whether PSS could predict subsequent segmental wall motion recovery.

Ninety-four patients (84 males, 10 females) presented with either STEMI or NSTEMI were enrolled. Serial echocardiography was performed shortly after PCI and at 3-, 6- and 12-month follow-up. During a follow-up period of 29.4 ±12.7 months, 22 patients (23.4%) had MACE and 17 patients had symptom-driven IRA revascularization. Using multivariate Cox regression analysis, the authors concluded that PSS measured at 3-month follow-up was an independent predictor for symptom-driven IRA revascularization and MACE. In addition, segmental wall motion recovery could be predicted by initial strain, strain rate and PSS.

Major issues:

1. Calculation errors noted in Table 1.

In the Event-free group, 32 out of 72 patients (44.4%) had LAD being the culprit vessel, whereas in the Event group 14 out of 22 patients (63.6%) had LAD being the culprit vessel. The data will need to be reanalyzed.

2. In Table 1, the number of patients with LM disease should be separately listed, since LM disease is critically important prognostically.

3. In this study the MACE rate for female was 50% and for male 20%. The proportion of females was significantly higher in the Event group than in the Event-free group (22.7% vs. 6.9%). This raises the question whether the between-group differences noted in LS and PSS were largely driven by the disproportionate number of female patients in each group. Given low enrollment of female patients in this study perhaps this confounding variable can be circumvented by excluding female patients from the study.

Minor issues:

Numerous typographical and grammatical errors will need to be corrected at revision.

6. PLOS authors have the option to publish the peer review history of their article (what does this mean?). If published, this will include your full peer review and any attached files.

Reviewer #1: **Yes: **Yijia Li

Reviewer #2: No

---

## [Author Response · Author response to Decision Letter 0]

25 Sep 2020

Reviewer #1: In this submission, the author tried to evaluate if the serial analysis of postsystolic shortening (PSS) by speckle tracking echocardiography (STE) could predict major adverse cardiovascular events (MACE), especially symptom-driven infarct-related artery (IRA) revascularization, and the segmental function improvement in post-myocardial infarction patients. They found that PSS is independently predictive for IRA revascularization and MACE. Initial PSS is also related to regional wall motion recovery. However, there are several issues that need to be discussed.

1. The author used PSS to evaluate the LV function for patients with newly-onset acute myocardial infarction. But it is not clear about how to measure/calculate the initial and after-admission PSS for the whole LV? Was it the average PSS for all 16 or 18 segments? Or is it the average PSS for all segments that showed postsystolic shortening? The former one may seem more reasonable. What is more, how about the time interval between AVC and PSS peak, which reflect the dyssynchrony of the LV and just like the standard deviation for time to peak of either strain or strain rate for all 16 and 18 LV segments. Time related parameters are also essential for evaluating the LV function after PCI and prognosis

Ans: PSS was the average PSS for all 18 segments. Sorry, we did not analyze time interval between AVC and PSS peak.

2. Other questions for PSS: the author mentioned that PSS at 3rd month is the independently predictive of IRA revascularization and MACE. And Seventeen patients received symptom-driven IRA revascularization at the median time of 7.7 months after acute myocardial infarction during the follow-up. (1) How about the time for MACE? (2) Dose all IRA revascularization after 3 months after admission? Because the result seemed to show that PSS can only indicate the IRA revascularization and MACE after 3 months after admission. Please discuss why only PSS at 3rd month is predictive of IRA revascularization and MACE?

Ans: (1) The median time for MACE was 7.5 months after acute myocardial infarction. (2) Two patients had IRA revascularization before 3 months after admission These events occurred at 11 and 60 days after admission individually. The other two MACE events before 3 months are two deaths at 38 and 78 days after admission. (3) we gave the potential explanation in “Discussion” paragraph as follows: Initial PSS was not predictive in the current study. Based on previous reports, the potential explanation is that PSS persisted even after reperfusion and is probably indicative of myocardial stunning or ischemic memory [10]. Thus, we presume that PSS at 3 months could be used to detect ongoing ischemia or restenosis rather than initial PSS.

2. The author used 16 segment model when calculated the WMSI and used 18 segment model to measure the strain and strain rate. Because the patients were enrolled from 2010 to 2013 and the 12 months follow-up echocardiography examination were ended at around the July 2014, and the echo data were all recorded, so why the author did not use 17 segment model for both WMSI and speckle tracking imaging (at least for longitudinal strain) when they analyzed the data now? They used the averaged strain and strain rate from all 18 segment, and how about the left ventricular bullseye plots? The global strain could be gotten from the bullseye plots.

Ans: It is a good recommend of using 17 segments model. In our study, two independent operators calculated WMSI and strain and strain rate. One used 16 segment model for WMSI. About strain and strain rate, we did not find the function of 18 or 17 segments model change from bullseye plot, initially. We calculated strain and strain rate and then exported to excel file for further analysis. Segmental strain and strain rate were recorded as 18 segment model.

3. It was a little bit hard to follow how patients had been treat and followed up. For example, when patients were admitted into the hospital, did all the patients had PCI? The author mentioned that PCI was performed as early as possible with high success rate, and information about diseased vessels were recorded, but they also mentioned that coronary angiographies were not done for all patients in limitation. The author did not mention when did they had the first echocardiography examination before “limitation”, and it should be put down in the “method”. Did every patient had been followed up for 12 months? In the method, the author said that patients were followed up for 12 months, but the results showed that the follow-up time is 29.4 ± 12.7 months. How about the dropout rate? A figure or a table to illustration the detail could be helpful.

Ans: All patients received PCI as early as possible after AMI was diagnosed. Routine following coronary angiography is not performed for all patients. Patients received coronary angiography only when he or she had angina symptoms. Flow chart for the study protocol is added to the manuscript. 

4. More details should be given when describing the angioplasty protocols, like did every patient underwent this procedure? How did the doctor decide when and whether the patient need the PCI? What is the definition about culprit vessel? Likewise, please provide the calculation formula for WMSI, and dose IRA revascularization include PCI and CABG?

Ans: Thank you. We gave more detail for angioplasty protocols in the method. All patients underwent PCI after acute myocardial infarction was diagnosed. IRA revascularization included 16 patient received PCI and 1 patient received CABG. 

5. How about the medicine use for enrolled patients when they admitted and during the follow-up? Did the hypertension and DM under control, because they might affect the MAC and IRA revascularization too.

Ans: Sorry, we did not have the records of the medicine during the follow up. We added this problem to limitation. Records for medicine at discharge were as follows: aspirin 92 (97.9%), clopidogrel 89 (94.7%), ACEI or ARB 41 (43.6%) and beta-blocker 50 (53.2%).

Minor comments:

1. Then author mentioned that they used the 2009 and 2015 recommendations of the American Society of Echocardiography and the European Association of Cardiovascular Imaging when preforming echo. Dose it means that they use 2009 guidelines when doing the conventional echocardiography analysis, while use the 2015 guidelines when measuring the STE parameter?

Ans Yes, the conventional echocardiography was performed in accordance with 2009 guideline and 2015 guideline was used for STE measurement.

2. Biochemical tests had been done immediately and at 8 hours and 16 hours later. Please make it clearer about the time point, like if it had been done 8 hours and 16 hours later after admission or PCI.

Ans: Only CK-MB had been done immediately and 8 and 16 hours after PCI. Other items of biochemical tests were done immediately at admission.

3. “The end of systole was defined as the first frame of the aortic valve in apical long-axis view, and each apical view or short-axis view was divided into 6 segments”. It should be “aortic valve close”. And “pule wave Doppler” should be better than “pulse wave velocity”.

Ans: Thank you

4. For SPSS 21.0, it should belong to IBM Corp (Released 2012. IBM SPSS Statistics for Windows, Version 21.0. Armonk, NY: IBM Corp).

Ans: Thank you

Reviewer #2: This study sought to determine, in patients presented with acute MI who were successfully treated with PCI, whether serial measurements of postsystolic shortening (PSS) by speckle tracking echocardiography could identify individuals at increased risk of developing major adverse cardiovascular events (MACE), and whether PSS could predict subsequent segmental wall motion recovery.

Ninety-four patients (84 males, 10 females) presented with either STEMI or NSTEMI were enrolled. Serial echocardiography was performed shortly after PCI and at 3-, 6- and 12-month follow-up. During a follow-up period of 29.4 ±12.7 months, 22 patients (23.4%) had MACE and 17 patients had symptom-driven IRA revascularization. Using multivariate Cox regression analysis, the authors concluded that PSS measured at 3-month follow-up was an independent predictor for symptom-driven IRA revascularization and MACE. In addition, segmental wall motion recovery could be predicted by initial strain, strain rate and PSS.

Major issues:

1. Calculation errors noted in Table 1.

In the Event-free group, 32 out of 72 patients (44.4%) had LAD being the culprit vessel, whereas in the Event group 14 out of 22 patients (63.6%) had LAD being the culprit vessel. The data will need to be reanalyzed.

Ans: Thank you. We corrected this error.

2. In Table 1, the number of patients with LM disease should be separately listed, since LM disease is critically important prognostically.

Ans: Thank you. We added new column for “LM”. In current study, patients with LM disease also have stenotic LAD.

3. In this study the MACE rate for female was 50% and for male 20%. The proportion of females was significantly higher in the Event group than in the Event-free group (22.7% vs. 6.9%). This raises the question whether the between-group differences noted in LS and PSS were largely driven by the disproportionate number of female patients in each group. Given low enrollment of female patients in this study perhaps this confounding variable can be circumvented by excluding female patients from the study.

Ans: Yes, this confounding variable can be circumvented by excluding female patients from the study. We will add this in the “limitation”. However, incidence of myocardial infarction in female patients is less than that in male. We did not exclude female patients because we want to show the real-word presentation. Because the number of events is are, it is difficult to do statistical analysis and the information could be insufficient if we excluded female patients. 

Minor issues:

Numerous typographical and grammatical errors will need to be corrected at revision.

Ans: We sent the manuscript to another professional company “Acceptediting” for English editing again. We did our great effort to correct typographical or grammatical errors at revision. Because there are numerous revisions, we did not mark it in the revised “manuscript” file.

---

## [Decision Letter · Decision Letter 1]

14 Oct 2020

PONE-D-20-11858R1

Usefulness of serial postsystolic shortening by speckle tracking echocardiography to predict major adverse cardiovascular events and segmental function improvement after acute myocardial infarction

PLOS ONE

Dear Dr. Hsu,

Thank you for submitting your manuscript to PLOS ONE. After careful consideration, we feel that it has merit but does not fully meet PLOS ONE’s publication criteria as it currently stands. Therefore, we invite you to submit a revised version of the manuscript that addresses the points raised during the review process.

Please address the remaining concerns of our Reviewers as noted below. 

We look forward to receiving your revised manuscript.

Kind regards,

John Lynn Jefferies, MD MPH FACC FAHA

Academic Editor

PLOS ONE

Reviewers' comments:

Reviewer's Responses to Questions

**Comments to the Author**

1. If the authors have adequately addressed your comments raised in a previous round of review and you feel that this manuscript is now acceptable for publication, you may indicate that here to bypass the “Comments to the Author” section, enter your conflict of interest statement in the “Confidential to Editor” section, and submit your "Accept" recommendation.

Reviewer #1: (No Response)

Reviewer #2: All comments have been addressed

2. Is the manuscript technically sound, and do the data support the conclusions?

Reviewer #1: Partly

Reviewer #2: Yes

3. Has the statistical analysis been performed appropriately and rigorously? 

Reviewer #1: N/A

Reviewer #2: Yes

4. Have the authors made all data underlying the findings in their manuscript fully available?

Reviewer #1: Yes

Reviewer #2: Yes

5. Is the manuscript presented in an intelligible fashion and written in standard English?

Reviewer #1: Yes

Reviewer #2: Yes

6. Review Comments to the Author

Reviewer #1: The author addressed some of the concerns in my review, but there are still some issues that need to be further discussed.

1. There is a parameter called time to peak, which should be provided with strain and strain rate by the software. It could make the manuscript more solid if the author adds that parameter as well.

2. In this study, 2 patients had IRA revascularization before 3 months after admission and the other 2 MACE events before 3 months. They did not have PSS at 3rd month. The author stated that PSS at 3rd month is independently predictive of IRA revascularization and MACE. These 5 patients should be excluded from the research.

3. Is it possible to reanalyzed the recorded echo imaging using the 17 segment model? According to my knowledge, EchoPAC provided a bullseye plot at least for longitudinal strain, and time to peak for strain and strain rate.

Reviewer #2: The authors did attempt to correct the calculation errors noted in Table 1. However, they have mistakenly placed the recalculated percentage values in the wrong column of the Table. Please revise.

(In the Event-free group, 32 out of 72 patients (44.4%) had LAD being the culprit vessel. In the Event group 14 out of 22 patients (63.6%) had LAD being the culprit vessel.)

7. PLOS authors have the option to publish the peer review history of their article (what does this mean?). If published, this will include your full peer review and any attached files.

Reviewer #1: **Yes: **Yijia Li

Reviewer #2: No

---

## [Author Response · Author response to Decision Letter 1]

2 Dec 2020

Reviewer #1: The author addressed some of the concerns in my review, but there are still some issues that need to be further discussed.

1. There is a parameter called time to peak, which should be provided with strain and strain rate by the software. It could make the manuscript more solid if the author adds that parameter as well.

Ans: Mechanical dispersion measured as standard deviation (SD) of time from onset Q in electrocardiography to peak negative longitudinal strain from 18 segments was added. 

2. In this study, 2 patients had IRA revascularization before 3 months after admission and the other 2 MACE events before 3 months. They did not have PSS at 3rd month. The author stated that PSS at 3rd month is independently predictive of IRA revascularization and MACE. These 5 patients should be excluded from the research.

Ans: We did not excluded these 5 patients because this initial study is designed to repeat echo studies including strain and strain rate in post-AMI patients to see if any parameter could predict MACE or IRA revascularization, not just focus on echo parameter at 3rd months. These 5 patients did not have PSS at 3rd month, and they are excluded when we calculated statistical analysis. 

3. Is it possible to reanalyzed the recorded echo imaging using the 17 segment model? According to my knowledge, EchoPAC provided a bullseye plot at least for longitudinal strain, and time to peak for strain and strain rate.

Ans: Sorry, we could not reanalyze the strain and strain rate using 17 segment model. Most raw data images were lost because the hard drive storage got broken and new version of EchoPAC was installed in Dec 2017.

Reviewer #2: The authors did attempt to correct the calculation errors noted in Table 1. However, they have mistakenly placed the recalculated percentage values in the wrong column of the Table. Please revise.

(In the Event-free group, 32 out of 72 patients (44.4%) had LAD being the culprit vessel. In the Event group 14 out of 22 patients (63.6%) had LAD being the culprit vessel.)

Ans: Sorry. I corrected it. Thank you.

---

## [Decision Letter · Decision Letter 2]

14 Dec 2020

Usefulness of serial post-systolic shortening by speckle tracking echocardiography to predict major adverse cardiovascular events and segmental function improvement after acute myocardial infarction

PONE-D-20-11858R2

Dear Dr. Hsu,

We’re pleased to inform you that your manuscript has been judged scientifically suitable for publication and will be formally accepted for publication once it meets all outstanding technical requirements.

Best regards,

John Lynn Jefferies, MD MPH FACC FAHA FHFSA

Academic Editor

PLOS ONE

Additional Editor Comments (optional):

Reviewers' comments:

Reviewer's Responses to Questions

**Comments to the Author**

1. If the authors have adequately addressed your comments raised in a previous round of review and you feel that this manuscript is now acceptable for publication, you may indicate that here to bypass the “Comments to the Author” section, enter your conflict of interest statement in the “Confidential to Editor” section, and submit your "Accept" recommendation.

Reviewer #1: All comments have been addressed

Reviewer #2: All comments have been addressed

2. Is the manuscript technically sound, and do the data support the conclusions?

Reviewer #1: Yes

Reviewer #2: Yes

3. Has the statistical analysis been performed appropriately and rigorously? 

Reviewer #1: Yes

Reviewer #2: Yes

4. Have the authors made all data underlying the findings in their manuscript fully available?

Reviewer #1: Yes

Reviewer #2: Yes

5. Is the manuscript presented in an intelligible fashion and written in standard English?

Reviewer #1: Yes

Reviewer #2: Yes

6. Review Comments to the Author

Reviewer #1: The authors addressed my concerns and my comments in the review. I recommend acceptance for publication.

Reviewer #2: (No Response)

7. PLOS authors have the option to publish the peer review history of their article (what does this mean?). If published, this will include your full peer review and any attached files.

Reviewer #1: **Yes: **Yijia Li

Reviewer #2: No

---

## [Editor Report · Acceptance letter]

21 Dec 2020

PONE-D-20-11858R2 

Usefulness of serial post-systolic shortening by speckle tracking echocardiography to predict major adverse cardiovascular events and segmental function improvement after acute myocardial infarction 

Dear Dr. Hsu:

I'm pleased to inform you that your manuscript has been deemed suitable for publication in PLOS ONE. Congratulations! Your manuscript is now with our production department. 

Kind regards, 

on behalf of

Dr. John Lynn Jefferies 

Academic Editor

PLOS ONE